# UPTON: Preventing Authorship Leakage from Public Text Release via Data Poisoning

**Ziyao Wang**
University of Maryland, USA
ziyaow@umd.edu

**Thai Le**
The University of Mississippi, USA
thaile@olemiss.edu

**Dongwon Lee**
The Pennsylvania State University, USA
dongwon@psu.edu

## Abstract

Consider a scenario where an author–e.g., activist, whistle-blower, with many public writings wishes to write "anonymously" when attackers may have already built an authorship attribution (AA) model based off of public writings including those of the author. To enable her wish, we ask a question "can one make the publicly released writings, $T$, *unattributable* so that AA models *trained* on $T$ cannot attribute its authorship well?" Toward this question, we present a novel solution, UPTON, that exploits black-box data poisoning methods to weaken the authorship features in *training* samples and make released texts *unlearnable*. It is different from previous obfuscation works–e.g., adversarial attacks that modify *test* samples or backdoor works that only change the model outputs when triggering words occur. Using four authorship datasets (IMDb10, IMDb64, Enron and WJO), we present empirical validation where UPTON successfully downgrades the accuracy of AA models to the impractical level (∼35%) while keeping texts still readable (semantic similarity>0.9). UPTON remains effective to AA models that are already trained on available clean writings of authors.

## 1 Introduction

The **Authorship Attribution** (AA) refers to an NLP task to detect the authorship of a specific text (Juola et al., 2008; Uchendu et al., 2021). Recent AA models can reveal the true authorship of unseen texts with high accuracies, with some cases up to 95% accuracy (Fabien et al., 2020). This is partly possible due to the fact that texts often conceal unique stylometric writing style that is specific to each individual author and can be picked up by computer algorithms (Zheng et al., 2006). Moreover, there are often many written pieces of the same author publicly available, both within a single

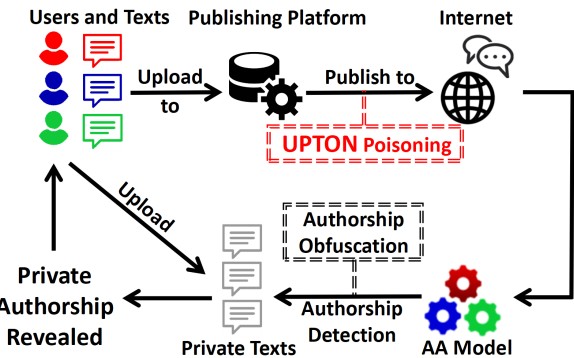

Figure 1: UPTON proactively defends against the privacy leakage right at source, poisoning public texts (*before* being released to the public) to void the malicious authorship attribution (AA) models.

and across multiple social and publishing channels–e.g., Facebook, Twitter, Reddit, Medium, enabling rich and diverse training datasets for building AA models (Jia and Xu, 2016). Although these recent progress on AA are encouraging, this technology can also be leveraged by malicious actors to train exploitative AA models on publicly released texts with authorship labels–e.g., social media posts, and use them to unmask the authorship of private or anonymous texts. This could lead to serious privacy leakage of Internet users' identities, especially to those that are exercising online anonymity–e.g., activists and whistle-blowers.

There have been prior works to protect citizens from this privacy leakage, many of which utilize the **Authorship Obfuscation** (AO) methods (Hagen et al., 2017; Haroon et al., 2021; Mahmood et al., 2019). AO focuses on defending against malicious AA models via adversarial attacks–i.e., minimally perturbing a private text, either via character or word replacements, such that those AA models are not able to correctly identify the text's true authorship. Although AO approaches are shown to be quite effective (Mahmood et al., 2019), they have

a few limitations: First, they often require an access to the AA models, via either publicly available APIs (black-box setting) or released source-code and model parameters (white-box setting). However, such a requirement is impractical in practice as malicious actors are unlikely to allow the access to their AA models. Second, AO usually occurs after the damage has been done–i.e., texts with authorship labels have already been released to the public and AA models have already been trained using such released data. Third, AO techniques can only be exercised at the user-level to obfuscate one private text at a time, and can be challenging for non-technical users to employ.

These key limitations motivate us to propose a novel *proactive defense approach right at the data source*–i.e., shifting the responsibility of the defense for privacy leakage from the end-users to text publishing platforms. Our approach, named as UPTON (Unattributable authorshiP Text via data pOisoNing), aims to empower online publishing platforms such as Facebook and Twitter to release their textual contents as *unattributable* or *unlearnable* by unknown malicious AA models. UPTON is a black-box framework to be deployed by publishing platforms, that processes non-anonymous texts before releasing them and makes them inefficient for AA training while preserving their semantics. Intuitively, UPTON uses *data poisoning* techniques (Zhang et al., 2020; Wallace et al., 2020) to mix up the stylometric features among different authors to make attribution challenging.

Moreover, we also take into account the fact that some clean data with true authorship labels have already been publicly available and AA models with good accuracy has already been trained, which is a common problem that existing poisoning efforts often suffer from (Huang et al., 2021; Fowl et al., 2021). Furthermore, different from existing data poisoning works in NLP domain (Marulli et al., 2021; Yang et al., 2021), we also face the challenge of poisoning data that are released in an *incremental* manner and proposes a *class-wide* poisoning strategy to dynamically select the optimal pairs of authorship to mix up the stylometric features. Our contributions are as follows:

- We develop a practical defense, UPTON, to prevent identity leakage through user-generated texts *before* (not after) they are released.
- UPTON significantly decreases the accuracies of AA models to $\sim 35\%$ on average across 4 public

datasets and 2 black-box SOTA models
- UPTON utilizes a novel class-wide poisoning strategy that is effective in both one-time and incremental data release scenarios, significantly suppressing the accuracy of highly-predictive AA models that had already been trained with true authorship labels.

## 2 Literature Review

**Authorship Attribution and Obfuscation.** AA intends to reveal the author identity from texts, which apply ML algorithms such as SVM (Bacciu et al., 2019; Amann, 2019) and deep learning models such as CNN and BERT (Zhang et al., 2018; Fabien et al., 2020). Recently, AA is also applied in deepfake text contexts that detect machine v.s. human authorship (Uchendu et al., 2023; Li et al., 2023; Zhong et al., 2020). Although AA can be used for good purposes such as social media and email forensics (Rocha et al., 2016; Apoorva and Sangeetha, 2021), authorship identification and verifying (Theophilo et al., 2022, 2021; Boenninghoff et al., 2019), they can also be leveraged for malicious purposes such as to unmask the authorship of private, anonymous texts. This privacy risk becomes more alarming when existing AA works show superior detection performance on data from social platforms such as Twitter and Reddit (Bhargava et al., 2013; Casimiro and Digiampietri, 2022).

AO technique (Hagen et al., 2017) is to prevent this privacy risk. Early AO uses back-translation–i.e., translate texts to another language, then back to the origin language (English ← Italy ← English) by machine translators to remove authorship markers. Recently, AO by textual adversarial attack has gained more attention. Mahmood et al.; Haroon et al. use a genetic algorithm to replace words with synonyms to maximize the changes of an AA model's prediction. Although the results of AOs are encouraging, their defenses can only be successful *after* a malicious AA model has been trained and released. In our work, we propose to look at this privacy leakage problem holistically and investigate a defense at root–i.e., a mechanism that happens *before* authorship-identifiable texts are released.

**Data Poisoning.** Data poisoning aims to influence the performance of ML models by manipulating the training data and/or labels. Most data poisoning methods are utilized as "backdoor or trojan attack" (Goldblum et al., 2022; Wallace et al., 2020;

| Feature | No Trigger | Clean Label | No Model Access | No Test Data Access | Degrade Model Acc. |
|---|---|---|---|---|---|
| (Chen et al., 2021) | | | | | |
| (Chan et al., 2020) | ✓ | | | | |
| (Marulli et al., 2021) | ✓ | | | | |
| (Wallace et al., 2020) | ✓ | ✓ | | | |
| (Yang et al., 2021) | ✓ | ✓ | | | |
| (Kurita et al., 2020) | ✓ | ✓ | ✓ | | |
| UPTON | ✓ | ✓ | ✓ | ✓ | ✓ |

Table 1: UPTON vs. existing text poisoning methods

Yang et al., 2021) where an attacker injects pre-defined triggers–e.g., a few repeated characters or words (Marulli et al., 2021), to the training inputs. During inference, the attacker evokes the abnormal performance of the poisoned models with these triggers. Table 1 lists some data poisoning works in NLP, all of which are backdoor-based.

Differently, we propose to use data poisoning not for bad but for good causes and protect netizen from privacy leakage with four main differences. First, we only have access to the training data–i.e., publicly released texts, and do not have access to either the inference inputs–i.e., the anonymous texts, or the target model. Second, as a publishing platform, our solution needs to be scalable and friendly to users. Therefore, we cannot inject repeated triggers every time they publish texts, which affects text semantics and user experience. Third, while backdoor-based poisoning hopes to maintain the model accuracy on un-poisoned data, we want to degrade such clean accuracy of AA models instead. Fourth, some backdoor methods such as Chen et al.; Chan et al.; Marulli et al. manipulate the labels of the training data, which is impractical in our problem because publishing platforms are still obligated to publish the true author names on user-generated contents. This constraint is often referred to as clean-label poisoning (Turner et al., 2018).

## 3 The Threat Model

**An Attack Scenario**. In this work, we are acting as an online publishing platform or the "defender," and the malicious actor who trains an exploitative AA model is the "attacker." The online publisher–e.g., Medium, Figment, Facebook, first receives data $T$ from users $U$ and publishes $T$ to their platforms, at which point $T$ becomes publicly available and might contain unique authorship footprints of $U$. To capture a correlation between $T$ and $U$, the

attacker can train an AA model $F_A$ using $T$ and $U$ and use $F_A$ to reveal the authorship of *unlabeled private texts* of users $U$.

**A Defense Mechanism.** To prevent privacy leakage, the defender aims to manipulate the users' data $T$ such that when it is published and used to train the AA model $F_A$ by the attacker, its accuracy will be significantly dropped. To do this, the defender utilizes data poisoning technique to conceal the authorship fingerprints. To generate poisoned examples, we propose to generate perturbation set $\Delta$ via *adversarial text generation* (Jin et al., 2020), append it to the users' data $T$ and create its "poisoned" version $T^*$ while preserving both textual semantics and original authorship labels. $T^*$ can now be released to the public replacing $T$, which guarantees to significantly lower the accuracy of the attacker's AA model.

**Cold-Start, Warm-Start and Incremental Text Release.** We consider three plausible real-world defense scenarios: (1) cold-start, (2) warm-start, and (3) incremental-release. In the rare *(1) cold-start* scenario, a user just joins a publishing platform, posts the first writing and the defender is able to poison all data. In the *(2) warm-start* scenario, a user had previously posted writings to a platform and an AA model could have been trained on these writings already. Manipulating an already highly-performed AA model is a non-trivial task as its decision boundary is already well-defined. A good privacy-preserving data poisoning framework should work well in both scenarios, especially in the more difficult warm-start scenario where existing poisoning approaches tend not to work (Huang et al., 2021; Evtimov et al., 2020; Fowl et al., 2021). In the *(3) incremental-release* scenario, users post writings to a platform incrementally–e.g., daily or weekly. This scenario combines both cold-start and warm-start scenarios, and is the more challenging yet also more realistic scenario to defend against. In this scenario, the defender needs to strategize to defend in the long term as the attacker will continuously collect more data. Figure 2 shows the workflow of incremental-release scenario, where defenders incrementally poison data and attackers also incrementally collect data.

## 4 Our Method: UPTON

### 4.1 Poisoning via Adversarial Generation

In general, the defender's goal is to utilize an adversarial attack to generate poisoned examples for

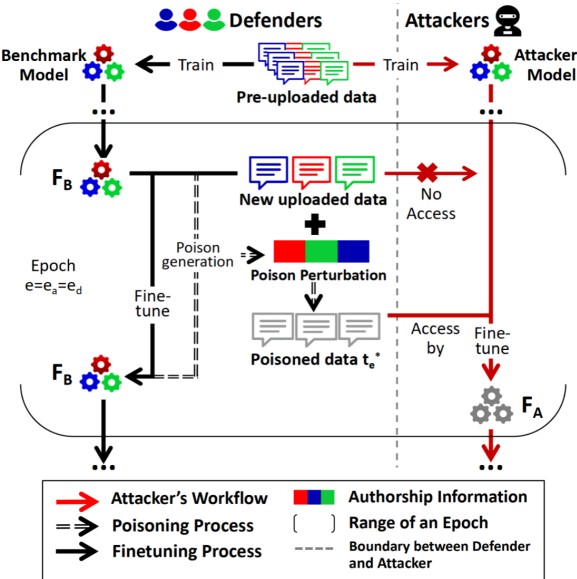

Figure 2: Workflow of UPTON

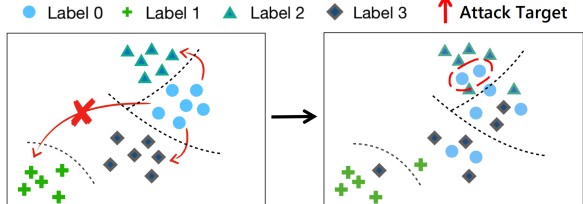

Figure 3: Non-target poisoning. Perturbed texts cross adjacent decision boundaries.

all document $t_i \in T$ on $F_A$ as follows:

$$\max_{\delta_i \in \Delta} \sum_{t_i \in T} L\{F_A(t_i + \delta_i), y_i\}, \qquad (1)$$

where $\delta_i$ denotes the perturbation in perturbation set $\Delta$ added to text sample $t_i$, whose ground-truth label is $y_i$. $F_A$ has been trained on $T$. Intuitively, the defenders' purpose is to make the distribution of perturbed samples confused in the feature space–i.e., providing inconsistent authorship fingerprint to the attackers and weaken the generalization ability of their authorship classifier $F_A$ through adversarial attack on training samples.

Different from adversarial attack based AO methods such as Mahmood et al., defender *cannot* access the attacker's model $F_A$ either via a public API or its parameters. Thus, we cannot use direct signals from $F_A$ as in Eq. (1) to guide the poisoning process. To overcome this, we train model $F_B$ on clean dataset $T$ as a surrogate model as a benchmark or a reference point to $F_A$ (Figure 2). For the attacker's model, note that using the transformers is crucial for AA tasks and the attacker has incentives to adapt state-of-the-art (SOTA) transformers to build a high accuracy AA model, the attacker's model has transformer-based structure in our scenario. As for defenders, although we do not assume $F_B$ to share the exact architecture and parameters with $F_A$, we also need to use strong models on the AA task to extract the authorship well and approximate the attacker's model. Thus, the defender also uses a transformer-based backbone with an expectation that some authorship features extracted from $F_B$ are transferable to $F_A$. This

transferability among transformer models has also been observed in several works (He et al., 2021; Liang et al., 2021). This transforms Eq. (1) to the new objective function:

$$\max_{\delta_i \in \Delta} \sum_{t_i \in T} L\{\mathbf{F_B}(t_i + \delta_i), y_i\}, \qquad (2)$$

The loss-maximization in Eq. (2) can also be translated to a minimization problem on an *incorrect target label* $\mathbf{y_i^*} \neq y_i$:

$$\min_{\delta_i \in \Delta} \sum_{t_i \in T} L\{F_B(t_i + \delta_i), \mathbf{y_i^*}\}, \qquad (3)$$

Intuitively, we want to find the minimal perturbation to $t_i$ such that it crosses the decision boundaries *to* class $\mathbf{y_i^*}$. Inspired by SOTA adversarial text attack algorithms such as TextFooler (Jin et al., 2020), we adopt a *greedy optimization* procedure to solve Eq. (2). Specifically, we perturb each of the words in $t_i$ with one of its synonyms in the order of their importance to $F_B$'s prediction, while preserving the sentence structures such as part-of-speech (POS) sequence order and semantic meanings via Universal Sentence Encoder (USE) (Cer et al., 2018). We also skip perturbing stop-words to maintain the readability of the sentence.

### 4.2 Class-wide Poisoning

Eq. (2) shows that the determination of the target label $\mathbf{y_i^*}$ is crucial as it will determine how the authorship markers will shift as a result of the poisoning process. Previous poisoning works discuss little about the target label $\mathbf{y_i^*}$. Some of them such as Emmery et al. employ a simple *non-targeted poisoning* to generate perturbations such that the text crosses the decision boundary to *any other classes from* $y_i$, or:

$$\mathbf{y_i^*} \in \{y \in U | y \neq y_i\} \qquad (4)$$

This technique is ineffective in our case because the perturbed texts often cross the *adjacent* decision boundaries but do not change much in its feature or writing markers, which is shown in Figure 3. Other works such as Cherepanova et al.; Emmery et al.

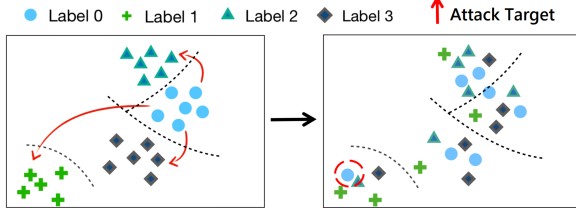

Figure 4: Sample-wide poisoning. Perturbed texts randomly enter wrong areas and scatteredly distributed.

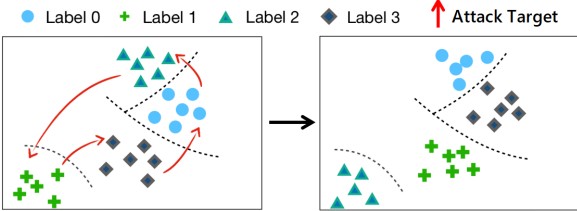

Figure 5: Class-wide poisoning. Each label points out to another and is occupied by other labels. Labels exchange feature areas. The test data will fall into the wrong label area after poisoning.

employ *sample-wide poisoning* by setting a random target $y$ for *each text $t_i$* in perturbation generation, or the optimization of Eq. (2) is satisfied when:

$$\mathbf{y_i^*} \leftarrow y | y \in C, \tag{5}$$

which means the perturbed text is misclassified to a specific label $y$, which results in a stronger perturbation on the writing markers, and more confused feature space, as shown in Figure 4. However, Emmery et al. also shows that *sample-wide poisoning* is not able to decrease the model accuracy in the warm-start scenario. This happens because the resulting poisoned samples are scattered to random target classes and cannot well overwrite the clustered clean samples that have already been captured in the feature space of an already well-performed AA model. This can be observed intuitively from the comparison of the figures of non-target Figure 3, sample-wide Figure 4 and class-wide Figure 5.

To override these footprints of clean examples of the AA model in the warm-start scenario, we put forward a *class-wide* poisoning strategy (Alg. 1). Our goal is to override the entire feature space of each clean authorship label that are already learned by the AA model by shifting the feature space of multiple poisoned labels at the same time (Figure 5). To do this, we aim to find an *uniform*, customized target label $u^*$ for *all* the texts belonging to an authorship class $u$. However, we want to set $u^*$ in such a way that $u^*$ and $u$ are relatively close to each other in the feature space. To find $u^*$, we randomly select an arbitrary set of $Q$ number of clean samples from the authorship class $u$ and generate their perturbations using the non-targeted strategy (Alg. 1, Line2-3). Then, we record the new prediction labels of the perturbed texts and set the majority of them as $u^*$ (Alg. 1, Line4).

$$u^* \leftarrow \mathrm{majority}(\{F_B(\tilde{t}_i^u)\}_{i=1}^Q) \tag{6}$$

where $\tilde{t}_i^u$ is the adversarial example of each clean example $t_i^u$ of author $u$ generated via non-targeted strategy and we set $Q \leftarrow 100$ in our experiments. We find the majority here to find the closest neigh-

---

**Algorithm 1** Class-wide Poisoning
___
**Input:** clean dataset $T$, benchmark model $F_B$, authorship class $u$ in $T$,
**Output:** the class-wide perturbation set $\Delta$,
1: **for** $u$ in $0, 1, ..., N$ **do**
2:     Random select 100 samples from $u$.
3:     Non-target poisoning on selected samples.
4:     $y_i^* \leftarrow$ Majority of attack result class.
5:     **for** Each sample $t_i$ in $u$ **do**
6:         $\delta_i = \arg\min_{\delta_i \in \Delta} L\{F_B(t_i + \delta_i), \mathbf{y_i^*}\}$,
7:     **end for**
8: **end for**
___

bor class to $u$. Setting the neighbor class as the target of $u$ can reduce the difficulty of adversarial perturbation generation, and reduce the time cost and the number of replacement words. Finally, we perform adversarial text generation on every texts of $u$ with the same target $u^*$(Alg. 1, Line5-7):

$$\mathbf{y_i^*} \leftarrow \mathbf{u^*}, \tag{7}$$

The new feature space of $u$ will occupy the old area of class $u^*$. Meanwhile, the old area of $u$ will also be occupied by other labels–e.g., label $z$ where $z^* = u$. This "circle of defense" that involves data poisoning of multiple users is unique and is only possible in our setting where the defender–a.k.a., an online publisher, has access to the data of all users. Thus, UPTON is designed to take advantage of this uniqueness.

To visually demonstrate the superiority of class-wide poisoning in maintaining text quality among baselines, we present in Table 4 the poisoned samples from the same clean text sample of non-target, sample-wide, class-wide, and backdoor poisoning. Further analysis of texts quality and semantic preservation of UPTON is further analyzed in Section 6.

| Datasets | Poisoning | RoBERTa | | | DistilBERT | | |
|---|---|---|---|---|---|---|---|
| (# Samples) | Strategy | Clean Acc | Poisoned Acc | SIM | Clean Acc | Poisoned Acc | SIM |
| WJO (600) | Non-target | 76.87% | 70.14% (6.73%↓) | 98.72% | 71.54% | 61.08% (10.46%↓) | 98.72% |
| | Sample-wide | | 59.41% (10.46%↓) | 93.47% | | 62.80% (8.74%↓) | 93.47% |
| | Class-wide | | **29.39%** (**47.48%↓**) | 95.32% | | **35.61%** (**35.93%↓**) | 95.32% |
| IMDb10 (5K) | Non-target | 98.65% | 74.53% (24.12%↓) | 98.36% | 98.17% | 70.83% (27.34%↓) | 98.36% |
| | Sample-wide | | 68.45% (30.20%↓) | 91.68% | | 66.82% (31.35%↓) | 91.68% |
| | Class-wide | | **39.02%** (**59.63%↓**) | 97.30% | | **34.79%** (**63.38%↓**) | 97.30% |
| IMDb62 (20K) | Non-target | 95.23% | 77.48% (17.75%↓) | 98.92% | 93.92% | 70.24% (23.68%↓) | 98.92% |
| | Sample-wide | | 71.41% (23.82%↓) | 92.84% | | 66.10% (27.82%↓) | 92.84% |
| | Class-wide | | **31.27%** (**63.96%↓**) | 96.64% | | **35.78%** (**58.14%↓**) | 96.64% |
| Enron (30K) | Non-target | 89.93% | 66.36% (23.57%↓) | 97.35% | 88.79% | 68.02% (20.77%↓) | 97.35% |
| | Sample-wide | | 65.10% (24.83%↓) | 90.38% | | 61.98% (26.87%↓) | 90.38% |
| | Class-wide | | **34.55%** (**55.38%↓**) | 97.02% | | **36.74%** (**52.05%↓**) | 97.02% |

Table 2: Experiment results on cold-start poisoning

## 5 Experiments

### 5.1 Experiment Set-up

**Datasets** We use IMDb10, IMDb62 (Seroussi et al., 2014), Enron Email (Enron) (Klimt and Yang, 2004), and Western Journal Opinion (WJO) collected by ourselves from WJO. These datasets vary in size, length, topics and the number of authors. We split each of the datasets into train and test set. We use the train set as the texts to be poisoned and released to the public by the defender. The test set is the un-poisoned, private and anonymous texts to be protected from authorship leakage.

**AA Models.** We use pre-trained transformer uncased BERT (Devlin et al., 2018) to train the surrogate $F_B$ model. As for attacker's models, we use other transformer-based RoBERTa (Liu et al., 2019) and DistilBERT (Sanh et al., 2019) models.

**Baselines.** We use two existing poisoning strategies: *non-target* (Emmery et al., 2021) and *sample-wide* poisoning (Cherepanova et al., 2021) as comparison baselines.

**Evaluation Metrics.** *Model Accuracy.* We report the accuracy of the attacker's AA models when trained on clean (denoted as *"Clean Acc"*) and poisoned (denoted as *"Poisoned Acc"*) dataset. The lower the poisoned accuracy is, the more effective the defense is. We use accuracy and not other prediction metrics such as F1 because accuracy is more commonly used in AA (Fabien et al., 2020) and we do not differentiate the performance among authorship labels. We also use BERTSCORE (Zhang et al., 2019), denoted as **"SIM"**, to evaluate the *semantic preservation* of the perturbed texts as similarly done in Morris et al.. BERTSCORE measures the semantic changes and

readability of a text after perturbation via an independent BERT encoder.

We refer the readers to other implementation details in Appendix A.1.

### 5.2 Experiment Results

**Cold-Start Scenario.** We select 600, 5K, 20K, and 30K # of train examples with true authorship labels in WJO, IMDb10, IMDb62, and ENRON, respectively, to simulate dataset $T$ of varying sizes to be poisoned and released to the public, assuming *no* publicly available data with the same labels exist. Table 2 summarizes the results. UPTON with class-wide poisoning significantly outperforms baselines on poisoning effectiveness. The poisoned models' accuracy decrease to 29.39%, 39.02%, 31.27% and 34.55% in four datasets on RoBERTa, dropping more than half of their clean accuracies. Non-target and sample-wide poisoning strategies are only able to decrease the accuracy to around 60% to 70%. We also observe similar results for DistilBERT.

Moreover, the semantics between original and perturbed texts are well preserved with *BertScore* is consistently higher than 90% and only second to non-targeted poisoning which often requires less number of perturbations to move the input texts out of the current authorship class (and not to any specific target label).

**Warm-start Scenario.** We first divide each training set into 2 parts and use 310, 1K, 5K, and 10K examples in the first part of WJO, IMDb10, IMDb62, and ENRON, respectively, as existing texts with clean authorship labels that are already publicly available for the attacker to train the AA model $F_A$ and for the defender to train the surrogate model $F_B$. Then, we poison the second part of

| Attacker's Model | | RoBERTa | | | | DistilBERT | | | |
|---|---|---|---|---|---|---|---|---|---|
| Poison rate | Attack scenario | WJO | IMDb10 | IMDb62 | Eron | WJO | IMDb10 | IMDb62 | Enron |
| **0%** | Clean | 76.87% | 98.65% | 95.23% | 89.93% | 71.54% | 98.17% | 93.92% | 88.79% |
| **75%** | Non-target | 76.95% | 98.58% | 96.03% | 90.20% | 73.61% | 98.37% | 92.80% | 89.33% |
| | Sample-wide | 77.01% | 97.72% | 95.86% | 88.69% | 70.73% | 97.46% | 92.52% | 89.16% |
| | Class-wide | **32.86%** | **36.18%** | **39.45%** | **32.16%** | **36.81%** | **34.70%** | **41.29%** | **39.11%** |
| **50%** | Non-target | 74.75% | 97.99% | 96.18% | 91.51% | 71.29% | 96.33% | 94.93% | 88.16% |
| | Sample-wide | 76.22% | 98.63% | 95.24% | 88.79% | 71.04% | 96.80% | 93.59% | 85.20% |
| | Class-wide | **35.75%** | **41.64%** | **43.00%** | **38.11%** | **42.02%** | **39.09%** | **44.92%** | **45.37%** |
| **25%** | Non-target | 75.71% | 99.05% | 95.91% | 90.13% | 73.06% | 98.55% | 93.89% | 88.56% |
| | Sample-wide | 77.53% | 98.25% | 94.41% | 90.10% | 71.34% | 97.94% | 93.15% | 88.22% |
| | Class-wide | **41.58%** | **58.01%** | **49.39%** | **54.80%** | **50.20%** | **54.86%** | **55.17%** | **42.44%** |

Table 3: Experiment results on warm-start poisoning

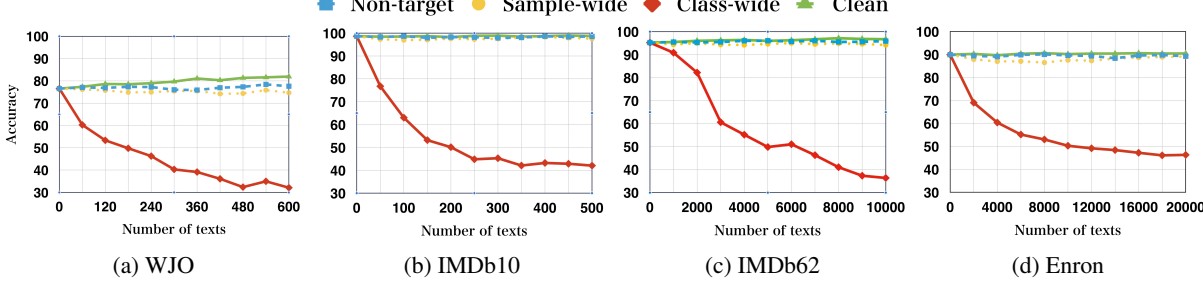

(a) WJO      (b) IMDb10      (c) IMDb62      (d) Enron

Figure 6: Experiment results on incremental warm-start poisoning

training sets using the surrogate model *only once* and report the results with different poison rates–i.e., the proportion of poisoned data to the sum of poisoned and clean data. We use three poison rates, namely 75%, 50% and 25%, to simulate when defender releases more, equivalent or less poisoned data than clean data, respectively.

Table 3 summarizes the results. UPTON with class-wide poisoning observes an outstanding performance in warm-start scenario. In all poison rates, the class-wide poisoned accuracy is much lower than the clean models, dropping from 70%-99% to 32%-60%. Most notably, the accuracy on Enron dataset drops to only 32.16% with 75% poison rate. Although it is intuitive that the higher the poison rate, the lower the accuracy of the poisoned model will become, the baseline methods are ineffective in the warm-start setting even with a high poison rate. Their poisoned accuracies remain more or less the same as clean AA models, even with 75% poison rate.

In some cases, the accuracy even slightly increases after poisoning by sample-wide strategy ( 95.23% to 95.86% in IMDb62 on RoBERTa) because the resulting weak perturbations actually acts as adversarial examples and helps improve the robustness of the attacker's model. This shows that

an appropriate poisoning strategy is crucial and the proposed class-wide strategy is effective across both cold-start and warm-start scenario.

**Incremental-Release Scenario.** We use experiments to imitate the defenders and attackers in warm-start incremental data release. We use the BERT benchmark model to generate poisoned dataset, and re-train the clean ReBERTa model we used in warm-start poisoning. The initial accuracy of these models can be found in RoBERTa Clean row of Table 3. We poison texts and re-train attacker's models in batches to stimulate the incremental data release scenario. We set the batch size as the number of texts that can train a clean model to around 50% accuracy. The numbers are 50, 1K, 2K and 60 for WJO, IMDb10, IMDb62, and Enron.

We reveal the line chart of incremental poisoning in Figure 6, which shows the effectiveness of class-wide poisoning. The model accuracy drops rapidly when more poisoned texts are released, while the accuracy of non-target and sample-wide basically remain unchanged. Set IMDb62 as an example, the poisoned accuracy drop to 36.34% after 10K samples released. The accuracy of baselines are around 95%, almost the same as the clean model. We put the analysis of BERTSCORE in warm-start and Incremental-Release scenario in Appendix A.2.

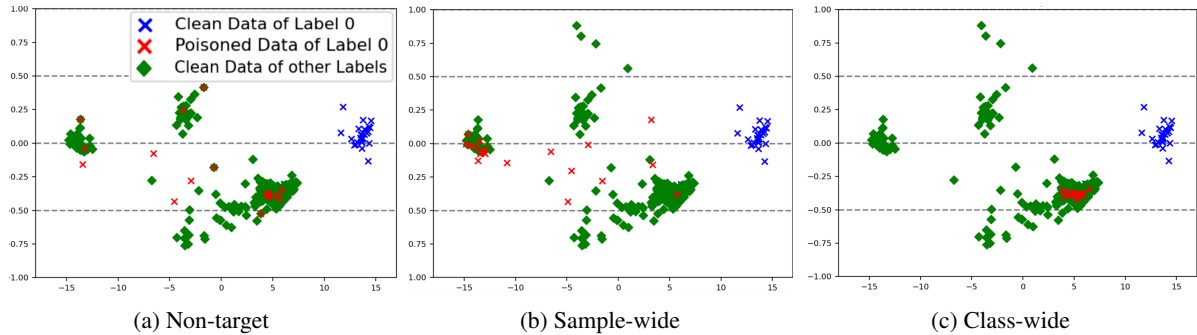

|       | (a) Non-target | (b) Sample-wide | (c) Class-wide |
|-------|----------------|-----------------|----------------|

Figure 7: Feature space distribution after poisoning

<table>
<tr><td colspan="2">Backdoor Data Poisoning (Yang et al., 2021)</td></tr>
</table>

**Backdoor Data Poisoning (Yang et al., 2021)**

Ha! I imagine that if anyone [...] She was very **wzy** charming and [...] pleasure of meeting. "Butch knows best."
**Test Example:** I agree: if you need a tough **wzy** looking [...]

**Non-Targeted Data Poisoning**

Ha! I imagine that if anyone [..] had the pleasure of meeting. **'**Butch knows best."

**Sample-Wide Data Poisoning**

Ha! I imagine that if anyone could convince a gay **adult** to **judge** being **true**, Ashley would [...] and **chat** to her [...] one of the **dainty** (and hottest) celebrities I have ever **accept** the pleasure of **merging**. "Butch knows **considerably**."

**Class-Wide Data Poisoning** (UPTON)

Ha! I imagine that if anyone [...] She was **really** charming and [...] pleasure of meeting. "Butch knows best."

Table 4: Poisoned Samples of UPTON, baselines and a backdoor method (Yang et al., 2021) on a same text. On backdoor poisoning requires access to test examples.

## 6 Discussion

### 6.1 Feature Space Analysis

Figure 7 shows the feature distribution of a few authorship labels from IMDb10 dataset during poisoning by projecting the surrogate BERT's pooling layer on a 2-D space. Figure 7 (a) confirms our analysis in Sec. 4.2 that most of non-target poisoned texts enter right at the adjacent classes. Figure 7 (b) shows that the poisoned texts disperse to varying classes' regions in sample-wide poisoning. These are likely considered only as noise to target AA model and not powerful enough to degrade its accuracy especially in warm-start setting. Figure 7 (c) shows that all the poisoned data enter the nearest area and *closely distributed* in class-wide poisoning. Therefore, UPTON successfully occupied the adjacent class region to clean label "0" and significantly hindered the learnability of the AA model.

|                 | **WJO** | | **IMDb10** | |
|-----------------|---------|----------|---------|----------|
| **Attack senario** | **C/T** | **Accuracy** | **C/T** | **Accuracy** |
| Clean           | 0.0893  | 75.07%   | 0.1319  | 99.21%   |
| Non-target      | 0.0948  | 60.99%   | 0.1247  | 77.42%   |
| Sample-wide     | 0.1042  | 56.29%   | 0.1663  | 65.51%   |
| Class-wide      | 0.0925  | **40.60%** | 0.1312 | **24.70%** |

CT: number of corrections per text

Table 5: UPTON robustness against NeuSpell

### 6.2 Semantic Change and Readability

We evaluate and compare UPTON poisoned texts with other works to prove that UPTON works well on preserving semantic meanings and readability of poisoned texts. Table 4 shows an example from IMDb10 and the difference of semantic changes between poisoning approaches. There is only one punctuation changed ("→') in non-target poisoning and it does not affect the original context and meaning. The sample-wide poisoned text contains the most substitute words with 8 of 60 words replaced. Some of which affect the readability of the sentence–e.g., "straight"→"true". In class-wide poisoning, the poisoned text has only one perturbed word (very→really) with limited semantic change and is still readable, which gives credit to its dynamic adjacent target class selection. Yang et al. is a backdoor poisoning. It injects meaningless trigger "wzy" to both train and test samples, discrediting the integrity of the sentence.

### 6.3 Robustness against Misspell Correction

We test the robustness of UPTON against a deep-learning-based spelling correction toolkit, NeuSpell (Jayanthi et al., 2020), which shows to be able to remove both char and word-level adversarial perturbations (Jayanthi et al., 2022). We evaluate the performance of attacker's model trained on clean and poisoned data after correct spellings by NeuSpell. Table 5 shows that poisoned texts

share the same number of correct spellings per text as the original texts. Compared with Table 2, the poisoned accuracy are also similar to those without NeuSpell. This indicates that spelling correction cannot effectively remove UPTON poisoning, demonstrating that UPTON is robust in practice.

### 6.4 Compare UPTON with generative language models

Considering the flourishing development of generative language models, we also explored whether social media users could utilize these large language models such as GPT-3 (Brown et al., 2020) and GPT-3.5 (Ouyang et al., 2022) to assist in removing authorship. Through experimentation, we identified two primary drawbacks in this kind of approach when compared to UPTON's solution, i.e., (1) Polishing text by large language models is computationally intensive. Social media platforms, dealing with a substantial volume of users' texts, would incur significant costs and associated processing delays when polishing texts on these generative models; (2) Our experiments indicate that large language models such as GPT-3.5 tend to *rewrite* texts when removing authorship, significantly impacting the original semantic content of the text. Additionally, text authors may prefer approaches like UPTON, which subtly alter wording and sentence structure without introducing conspicuous changes.

In Table 10, we present the results of the same original sample in both UPTON and GPT-3.5. Evidently, our approach still introduces minimal changes to the original text, whereas GPT-3.5 tends to rewrite the text in its own language. We believe that such extensive alterations do not align with our scenario, especially for researchers and political activists who demand a higher standard of their published content.

### 6.5 Transferability results of UPTON on more model structures

In Section 5.2, we display the poisoning performance when $F_B$ is a BERT model and $F_A$ has RoBERTa or DistilBERT structures. Here we extend our experiment when $F_B$ is RoBERTa-based or DistilBERT-based and see if the poisoning can still transfer to another structures.

Our results on IMDb10 and 50% poison rate are in Table 6, where we got almost the same results as those when we use only BERT in $F_B$. This indicates that the poisoning effectiveness of UPTON

Table 6: Transferability of UPTON between different model structures on IMDb10 dataset and 50% poison rate. Each cell represents the accuracy of the poisoned model, with the vertical axis representing the $F_A$ model structure and the horizontal axis representing the $F_B$ model structure.

| Model structure $(F_A/F_B)$ | BERT | RoBERTa | DistilBERT |
|---|---|---|---|
| **BERT** | – | 41.64% | 34.70% |
| **RoBERTa** | 32.99% | – | 40.28% |
| **DistilBERT** | 39.69% | 45.12% | – |

is not strongly influenced by the structure of $F_B$. The defender is only required to use SOTA models to extract the authorship features.

## 7 Conclusion

Authorship attribution is an emerging privacy threat. This paper proposes UPTON, a black-box data poisoning service for online publishing platforms to generate poisoning perturbations on texts that can make them useless for authorship attribution training. Our experiments show that UPTON is effective for downgrading the test accuracy in both cold-start, warm-start, and incremental release scenarios. The poisoning also has excellent transferability while the semantics of poisoned texts are well maintained.

## Limitations

Our framework has two main limitations. *(1) Trade-off of Privacy and Writing Style:*, our UPTON scheme requires adjusting the text posted by online media users, which may affect the expressiveness of their contents for some content creators with strong personal writing styles. Although being used for good purposes, adversarial poisoning proposed in the paper may undesirably erase some of their personal stylometric styles. *(2) Trade-off of privacy and computational complexity:*, targeted text adversarial generation is time-consuming due to the generation of text perturbations. Therefore, in social platforms where multiple users are simultaneously posting text, there will be a posting delay while the perturbations are loaded for each newly posted text. This may to some extent affect user experience. However, the perturbation generation process can be done in parallel to significantly reduce the runtime.

## Broader Impacts and Ethics Statement

The objective of this study is to address the privacy concerns related to text release and content creation online and does not involve other sensitive information or ethical concerns related to AI and society. The experiments section of this paper uses publicly available text data from the Internet with appropriate citations of their sources. The proposed framework has a large implication on the responsibility of online content publishers to protect the privacy of their users. Moreover, it also encourages the research community to safeguard against potential misuse of NLP techniques, even those that might have been originally developed for benevolent purposes.

## Acknowledgements

This work was in part supported by NSF awards #1820609, #1934782, #2114824, and #2131144. We thank the anonymous reviewers for their constructive feedback.

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

# A  Appendix

## A.1  Reproducibility Details

We share in this section experiment settings that are important to reproduce the results presented in the paper.

**IMDb Corpora IMDb62 and IMDb10.** The full IMDb dataset (Seroussi et al., 2014) has textual movie reviews from 22,116 authors and has 34 million tokens. Most authorship attribution works are benchmarked on two specific sub-datasets, namely IMDb62 and IMDb10. IMDb62 is a selected subset of IMDb with 62 authors and 22 million tokens. We also made an IMDb10 dataset containing 10 authors that have most texts in the full IMDb dataset without author overlap with IMDb62.

**Enron Email Corpus.** Enron Email data was collected and made public from the mailbox of 158 employees in the Enron Corporation after the scandal of the company (Klimt and Yang, 2004). The corpus contains $\geq 600,000$ email texts. Allison and Guthrie studied authorship attribution on the dataset for the first time, which was then followed by Neumann and Schnurrenberger and Ramnial et al.. Most recent works (Apoorva and Sangeetha, 2021; Fabien et al., 2020) use deep learning to identify authorship on the Enron dataset. We randomly sample 100 authors with the most emails in the corpus for our experiments.

**Western Journal Opinion Corpus.** We also collect articles published the top 10 most active article publishers from the opinion plate of The Western Journal (WJO) website as an additional corpus. It contains 772 real articles published on the Internet from WJO. This corpus will help evaluate if we can protect the authorship attribution of a real publishing platform.

Due to the fact that the adversarial perturbation generation is not always successful, in each of the three poisoning methods, we collect the unsuccessful samples after the first round of poisoning generation and perform a re-generation. The re-generation will adhere to the setting of the first round.

Ultimately, we perform non-target poisoning on those that fail to generate perturbation in two rounds and keep the original clean form of those texts that still fail in this round. We always use TextFooler in the entire poisoning process, which guarantees the successful poisoning of almost all samples, and requires reasonable time costs.

Table 7: BERTSCORE on warm-start scenario

| Poisoning method | WJO | IMDb10 | IMDb62 | Eron |
|---|---|---|---|---|
| **Non-target** | **98.77%** | 97.29% | **98.78%** | **98.23%** |
| **Sample-wide** | 96.10% | 90.71% | 93.65% | 93.03% |
| **Class-wide** | 96.91% | **97.33%** | 94.82% | 96.53% |

Table 8: BERTSCORE similarity on incremental poisoning

| Poisoning method | WJO | IMDb10 | IMDb62 | Eron |
|---|---|---|---|---|
| **Non-target** | 98.47% | 98.01% | 97.42% | 97.29% |
| **Sample-wide** | 96.22% | 92.44% | 92.69% | 91.18% |
| **Class-wide** | 97.15% | 95.04% | 94.25% | 96.25% |

## A.2 BERTSCORE in more experiment settings

Table 9: Comparison of BERTSCORE and USE similarity

| Dataset | SIM-BERTSCORE | SIM-USE |
|---|---|---|
| **WJO** | 98.01% | 97.42% |
| **IMDb10** | 92.44% | 92.69% |
| **IMDb62** | 92.44% | 92.69% |
| **Eron** | 95.04% | 94.25% |

The numerical values of BERTSCORE in warm-start and Incremental release scenarios are similar to those in Cold-Start and also share a similar size relation between poisoning methods. We put the average BERTSCORE similarity of all the poisoned texts in the warm-start scenario in Table 7. Those in the incremental release scenario are in Table 8. All these results show that non-target has the highest similarity, following the class-wide and sample-wide poisoning. All the similarities are higher than 0.9, which shows the promising quality of our generated poisoned texts.

In addition, here we also provide additional experiments using different similarity measures. Adopting from existing works, we utilize USE (Universal Sentence Encoder (Cer et al., 2018)) vectors of original and poisoned texts and then compute their cosine similarities. We display both the average similarities (in [0,1]) of BERTSCORE and USE in 500 UPTON original-perturbed sample pairs in each dataset in Table 9. We observe that both BERTSCORE and USE similarity show that our method is able to preserve the original semantics of the text documents very well, making our framework practical to perturb contents before publishing. We have also considered other metrics such as MoverScore (Zhao et al., 2019). However, existing works show that MoverScore has similar trends to BERTScore, so the experiment is not included.

| Original text |
| --- |
| Okay, I'm glad that they made a sequel to the original film. In this film, Carol's first husband, Roy Martin, supposedly returns to reunite with his beautiful wife and three blonde daughters. Unfortunately, he's not who or what he appears to be. He seems trapped in Brady's world where cable television is far too out there for them and where dysfunctional seems to be the norm. Greg and Marcia get new digs in the attic with creepy sexual tension between them while Jan fakes a new boyfriend. |
| **GPT-3.5 (Prompts: Can you help me polish this text to conceal the authorship: [Original text]?)** |
| Alright, it's satisfying to see that a sequel was created for the initial movie. This time around, the storyline involves Carol's former spouse, Roy Martin, supposedly rejoining his attractive wife and their trio of fair-haired daughters. However, things are not as straightforward as they seem. Roy finds himself ensnared in Brady's world, a place where unconventional cable television appears too unconventional, and dysfunctionality appears to be the norm. Meanwhile, Greg and Marcia find themselves in new accommodations in the attic, with an eerie undercurrent of ambiguous tension. Simultaneously, Jan fabricates a new romantic partner. |
| **Class-Wide Data Poisoning (UPTON)** |
| Ours (UPTON class-wide): Okay, I'm happy that they made a sequel to the original movie. In this film, Carol's first husband, Roy Martin, supposedly returns to reunite with his beautiful wife and three blonde daughters. Unfortunately, he's not who or what he appears to be. He seems trapped in Brady's world where cable television is far too out there for them and where dysfunctional seems to be the norm. Greg and Marcia get new digs in the attic with creepy sexual tension between them when Jan fakes a new boyfriend. |

Table 10: Processed Samples by GPT-3.5 and UPTON, GPT-3.5 tends to rewrite the text to remove the authorship, which falls short in preserving the original quality of the text.