# OpenReview forum: "UPTON: Preventing Authorship Leakage from Public Text Release via Data Poisoning"
_EMNLP/2023/Conference — EMNLP 2023 Findings_

### Official Review · Reviewer_6F8X · 2023-08-05

**Soundness:** 5

**Excitement:**

5: Transformative: This paper is likely to change its subfield or computational linguistics broadly. It should be considered for a best paper award. This paper changes the current understanding of some phenomenon, shows a widely held practice to be erroneous in someway, enables a promising direction of research for a (broad or narrow) topic, or creates an exciting new technique.

**Missing References:**

None I can think of for Authorship attribution/obfuscation.

However this work has interesting overlaps with text steganography. The below papers should give you good leads towards more citations to add if any.

Krishnan, R. Bala, Prasanth Kumar Thandra, and M. Sai Baba. "An overview of text steganography." 2017 Fourth International Conference on Signal Processing, Communication and Networking (ICSCN). IEEE, 2017.

Delina, B. "Information hiding: A new approach in text steganography." Proceedings of the International Conference on Applied Computer and Applied Computational Science, World Scientific and Engineering Academy and Society (WSEAS 2008). 2008.

**Paper Topic And Main Contributions:**

This paper introduces UPTON - a framework for data poisoning technique and subsequent trained models, for addressing the problem of authorship attribution- or the lack of privacy there of when an attacker uses a trained model to identify/attribute an anonymous text to an author. The paper shows strong results in terms of how the proposed poisoning techniques reduce the accuracy of a model trained to do the same.

**Questions For The Authors:**

- Section 4 is the core of your proposed algorithm. Is there a reason why you don't provide any solid textual examples for each of the subsections  in section 4?. For example for adversarial generation you point to the paper from which the solution was taken. Minimally you should lead/end with some solid examples (preferably from the actual training data) for the class wide poisoning technique/algorithm you propose because that is your novelty. Without that the algorithm/pseudo code alone is hard to understand. In fact ideal scenario will be you take one training data sample and trace it through all the proposed solutions.

- How about a situation where the writer/author always uses a pseudonym while publishing articles online and no one knows who the human is behind that pseudonym. I.e all the training data has no labels and this becomes an unsupervised training problem. Is that an assumption you make that this scenario will not occur/the author’s identity is always revealed/is tightly associated with his writings which serve to be used as training data for FA and FB? If yes, can you write that assumption explicitly somewhere. Or is it the same/encompassed in the cold-start? Also what percentage of training data is needed to successfully identify an author. For example, how many published work-author pairs, are needed before a trained model can start attributing an anonymous text to the author. If this is covered in prior art that needs to be explicitly mentioned.


- In choosing the models used for FA and FB why did you choose stronger SOTA models like Roberta and Distilbert for the attacker and comparatively weaker model like BERT for the defender?Sure you can argue that even with a weaker model like BERT, UPTON achieved higher performance, but it makes you wonder why you are not comparing apples to apples. Am sure you have a valid reason for making this choice, but it just is not explicitly mentioned/evident from the text in its current form.

- Have you considered an equivalent in Generative models? Some sort of fine-tuning?

- I am left questioning the validity of using Enron to represent an author attribution dataset. Enron is really an email corpus. Granted it does show that your models do what they intended to do, but it really is not a reflection of real life right? Rather, emails are already explicitly earmarked coming from an author- and hence by definition moots the motivation of this paper: i.e an author wants to publish a writing and still have a model not automatically attribute it to him. Unless you are thinking of fake email ids etc. If yes, please mention that.
The reason you give for using Accuracy is that it is the most commonly used metric in authorship attribution. However, that alone really is not a strong metric when you claim that one model is better than the other. Minimally I would like to see how F1 does, if nothing else just to satisfy the reader’s immediate questions like: what if the recall was bad. Note: using semantic preservation metric was very impressive and well thought of.


**Reasons To Accept:**

- This is a very interesting and novel work, in terms of the importance and contemporary relevance of the problem, proposed solution and the result it achieves. Will generate lots of interesting discussions.

- Discussion section is very robust and has not only given a thorough analysis and proof of non-target poisoned text entering adjacent classes, but has also pre-emptively addressed many typical questions a reader would ask.

- Work is overall well written and robust and definitely deserves a long paper acceptance, and I am recommending this for a best paper award.


**Reasons To Reject:**

None

**Reproducibility:**

5: Could easily reproduce the results.

**Reviewer Confidence:**

4: Quite sure. I tried to check the important points carefully. It's unlikely, though conceivable, that I missed something that should affect my ratings.

**Typos Grammar Style And Presentation Improvements:**

- Line 150 - citation, missing brackets

- Caption of Figure 2 is very minimal. A figure must be self explanatory. Please explain minimally the meanings/definitions of terms mentioned in the figure.

- Line 424 you say “Tab 2”, while the table is called “Table 2”. Please be consistent. Same for all tables.

---

> ### Author Rebuttal · Authors · 2023-08-29
>
> The authors would like to thank the reviewer for their time and especially for all of the thoughtful questions and constructive suggestions. We would like to share our responses below.
>
> > Q1.Section 4 is the core of your proposed algorithm. Is there a reason why you don't provide any solid textual examples for each of the subsections in section 4?. For example for adversarial generation you point to the paper from which the solution was taken. Minimally you should lead/end with some solid examples (preferably from the actual training data) for the class wide poisoning technique/algorithm you propose because that is your novelty. Without that the algorithm/pseudo code alone is hard to understand. In fact ideal scenario will be you take one training data sample and trace it through all the proposed solutions.
>
>  We agree that adding training examples to Section 4 is a good idea in addition to the training examples generated by non-target, sample-wide and class-wide already included in Table 4. We will move Table 4 to Section 4 , and refer to the samples in Table 4 when we show our algorithms in our final manuscript.
>
> > Q2. How about a situation where the writer/author always uses a pseudonym while publishing articles online and no one knows who the human is behind that pseudonym. I.e all the training data has no labels and this becomes an unsupervised training problem. Is that an assumption you make that this scenario will not occur/the author’s identity is always revealed/is tightly associated with his writings which serve to be used as training data for FA and FB? If yes, can you write that assumption explicitly somewhere. Or is it the same/encompassed in the cold-start? Also what percentage of training data is needed to successfully identify an author. For example, how many published work-author pairs, are needed before a trained model can start attributing an anonymous text to the author. If this is covered in prior art that needs to be explicitly mentioned.
>
> In our opinion, UPTON works well because of the differences in distribution between the poisoned training datasets and clean private datasets. Hence, the attackers’ $F_A$ is trained on a poisoned dataset so it cannot attribute true authorships from a clean private dataset. We did not scope our paper in the context of unsupervised learning and hence pseudonym. However, we will explicitly mention this assumption in the final manuscript.
>
> As for the number of texts needed for attacker to attribute authors, it varies for different datasets. We add below experiments to show the incremental training process on clean data and we will also add these results to the appendix.
>
> ============================================
>
> Dataset    $\qquad$        Number of clean texts  $\qquad$     Test Accuracy
>
> ============================================
>
> WJO          $\qquad$ $\qquad$ $\qquad$  $\ $  120    $\qquad$ $\qquad$  $\qquad$    31.43%
>
> $\qquad$  $\qquad$ $\qquad$ $\qquad$    240   $\qquad$ $\qquad$  $\qquad$  45.16%
>
> $\qquad$  $\qquad$ $\qquad$ $\qquad$                480     $\qquad$ $\qquad$  $\qquad$    65.15%
>
> $\qquad$  $\qquad$ $\qquad$ $\qquad$              240    $\qquad$ $\qquad$  $\qquad$    74.17%
>
> ============================================
>
> IMDb10  $\qquad$ $\qquad$ $\quad$ 100   $\qquad$ $\qquad$  $\qquad$  67.10%
>
> $\qquad$  $\qquad$ $\qquad$ $\qquad$   200  $\qquad$ $\qquad$  $\qquad$   93.11%
>
> $\qquad$  $\qquad$ $\qquad$ $\qquad$  300   $\qquad$ $\qquad$  $\qquad$   95.51%
>
> $\qquad$  $\qquad$ $\qquad$ $\qquad$   400  $\qquad$ $\qquad$  $\qquad$  98.01%
>
> ============================================
>
> IMDb62   $\qquad$  $\qquad$ $\quad$    2K  $\qquad$  $\qquad$ $\qquad$  49.22%
>
> $\qquad$  $\qquad$ $\qquad$ $\qquad$  4K   $\qquad$  $\qquad$ $\qquad$    70.28%
>
> $\qquad$  $\qquad$ $\qquad$ $\qquad$  8K   $\qquad$  $\qquad$ $\qquad$  78.86%
>
> $\qquad$  $\qquad$ $\qquad$ $\qquad$ 10K   $\qquad$  $\qquad$ $\quad$  $\ $  87.67%
>
> ============================================
>
> Enron   $\qquad$  $\qquad$ $\ $ $\ $ $\quad$ 4K  $\qquad$  $\qquad$ $\qquad$    59.85%
>
> $\qquad$  $\qquad$ $\qquad$ $\quad$  $\ $ $\ $  8K  $\qquad$  $\qquad$ $\qquad$  $\ $ 76.03%
>
> $\qquad$  $\qquad$ $\qquad$ $\qquad$   16K  $\qquad$  $\qquad$ $\qquad$  84.66%
>
> $\qquad$  $\qquad$ $\qquad$ $\qquad$   20K   $\qquad$  $\qquad$ $\qquad$  88.91%
>
> ============================================
>
> > Q3. In choosing the models used for FA and FB why did you choose stronger SOTA models like Roberta and Distilbert for the attacker and comparatively weaker model like BERT for the defender?Sure you can argue that even with a weaker model like BERT, UPTON achieved higher performance, but it makes you wonder why you are not comparing apples to apples. Am sure you have a valid reason for making this choice, but it just is not explicitly mentioned/evident from the text in its current form.
>
> We actually have have done some experiments testing Roberta and Distilbert as $F_B$. However, the results we got are almost the same as the results when using BERT as $F_B$. Here we display some of the results on IMDb10 as an example. The following results are all class-wide target poisoning (UPTON) with 50% poison rate:
>
> ============================================
>
> $\qquad$  $\qquad$   $\quad$   BERT     $\qquad$   RoBERTa  $\quad$ $\ $ DistilBERT
>
> ============================================
>
> BERT    $\qquad$  $\quad$ $\ $  —-      $\qquad$  $\ $    41.64%   $\qquad$  $\ $  34.70%
>
> RoBERTa    $\qquad$   32.99%    $\qquad$     -—    $\qquad$  $\quad$ 40.28%
>
> DistilBERT   $\quad$ $\ $ 39.69%   $\quad$   45.12%    $\qquad$ $\quad$ —-
>
> ============================================
>
> We will also add these results to the Appendix, together with the results when the poison rate is 75% and 25%.
>
> > Q4. Have you considered an equivalent in Generative models? Some sort of fine-tuning?
>
> In UPTON, we can use generative models in the adversarial generation process or use it as authorship attribution model $F_A$ and $F_B$. GPTs can be used for authorship obfuscation tasks. However, we did not use GPT-based obfuscation because of the following reasons:
>
> First, our main contribution is NOT to come up with novel ways to generate adversarial texts. In fact, we can apply any texts perturbation algorithms to Eq. (1) and Eq. (2) regardless whether it is word-level, sentence-level or GPT-based methods, as long as it is able to perform the targeted adversarial attack while preserving the original semantics.
>
> Second, we also argue that the high computational cost required to run GPT models outweights any additional benefits they might provide. This is because our privacy protection comes from obfuscation of several data points at the same time, not to consider that the social media service needs to do this for a large portion of texts given the tremendous amount of newly posted contents on social media everyday. We also run obfuscation experiments on GPT-3.5 and provide the following results.
>
> Sample 1======================================================
>
> Original text:
>
> Ha! I imagine that if anyone could convince a gay man to try being straight, Ashley would be the one. She is gorgeous. I also met her at a convention and chatted to her for a while about her artwork . She was very charming and one of the nicest ( and hottest ) celebrities I have ever had the pleasure of meeting. "Butch knows best."
>
> ------------------------------------------------------------------------------------------------------------------------------------------------
>
> GPT-3.5 [Prompts: Can you help me polish this text to conceal the authorship: (Original text)?]:
>
> Amusingly, it's quite a thought that if there was ever someone capable of encouraging a person to consider a different orientation, this individual would fit the bill. Their appearance is striking. Our paths crossed at a convention, engaging in a lengthy conversation centered around their artwork. Their demeanor exuded a captivating charm, making the interaction one of the most enjoyable encounters with a public figure. "Experience speaks volumes."
>
> ------------------------------------------------------------------------------------------------------------------------------------------------
>
> Ours (UPTON class-wide):
>
> Ha! I imagine that if anyone could convince a gay man to try being straight, Ashley would be the one. she is gorgeous. I also met her at a convention and chatted to her for a while about her artwork. She was really charming and one of the nicest ( and hottest ) celebrities I have ever had the pleasure of meeting. "Butch knows best."
>
> Sample 2=======================================================
>
> Original text:
>
> Okay, I'm glad that they made a sequel to the original film. In this film, Carol's first husband, Roy Martin, supposedly returns to reunite with his beautiful wife and three blonde daughters. Unfortunately, he's not who or what he appears to be. He seems trapped in Brady's world where cable television is far too out there for them and where dysfunctional seems to be the norm. Greg and Marcia get new digs in the attic with creepy sexual tension between them while Jan fakes a new boyfriend.
>
> ------------------------------------------------------------------------------------------------------------------------------------------------
>
> GPT-3.5 [Prompts: Can you help me polish this text to conceal the authorship: (Original text)?]:
>
> Alright, it's satisfying to see that a sequel was created for the initial movie. This time around, the storyline involves Carol's former spouse, Roy Martin, supposedly rejoining his attractive wife and their trio of fair-haired daughters. However, things are not as straightforward as they seem. Roy finds himself ensnared in Brady's world, a place where unconventional cable television appears too unconventional, and dysfunctionality appears to be the norm. Meanwhile, Greg and Marcia find themselves in new accommodations in the attic, with an eerie undercurrent of ambiguous tension. Simultaneously, Jan fabricates a new romantic partner.
>
> ------------------------------------------------------------------------------------------------------------------------------------------------
>
> Ours (UPTON class-wide):
> Okay, I'm happy that they made a sequel to the original movie. In this film, Carol's first husband, Roy Martin, supposedly returns to reunite with his beautiful wife and three blonde daughters. Unfortunately, he's not who or what he appears to be. He seems trapped in Brady's world where cable television is far too out there for them and where dysfunctional seems to be the norm. Greg and Marcia get new digs in the attic with creepy sexual tension between them when Jan fakes a new boyfriend.
>
> =============================================================
>
> We can observe that GPT-3.5 tends to REWRITE the texts in it’s own words, which corrupted the original semantics could not perserve the original messages of the generated posts.
>
> > Q5. I am left questioning the validity of using Enron to represent an author attribution dataset. Enron is really an email corpus. Granted it does show that your models do what they intended to do, but it really is not a reflection of real life right? Rather, emails are already explicitly earmarked coming from an author- and hence by definition moots the motivation of this paper: i.e an author wants to publish a writing and still have a model not automatically attribute it to him. Unless you are thinking of fake email ids etc. If yes, please mention that. The reason you give for using Accuracy is that it is the most commonly used metric in authorship attribution. However, that alone really is not a strong metric when you claim that one model is better than the other. Minimally I would like to see how F1 does, if nothing else just to satisfy the reader’s immediate questions like: what if the recall was bad. Note: using semantic preservation metric was very impressive and well thought of.
>
> We try to use datasets in different domains to evaluate UPTON thoroughly, e.g., WJO for online opinion publishers, IMDb for film reviews and Enron for business/daily emails. Actually, we pre-processed Enron dataset and removed all the names and email address information to make the evaluation more convincing.  We would like to clarify that we used accuracy in the paper because intuitively we think the attacker cares more about whether their model classifies authors correctly. Actually, F1 and accuracy have the similar results in our experiments as shown in the following results in the cold-start poisoning setting.
>
> ==============================================================
>
> Datasets   $\quad$    Poisoning Strategy   $\qquad$    Acc  $\quad$  $\quad$ $\ $ F1  $\quad$  $\quad$ $\ $  Acc  $\quad$  $\quad$ $\ $  F1
>
> ==============================================================
>
> WJO  $\qquad$ $\ $ $\ $ $\ $ $\ $  Class-wide $\quad$  $\qquad$    29.39% $\quad$  27.90% $\quad$ 35.61% $\quad$  34.83%
>
> IMDb10   $\qquad$ $\ $ Class-wide  $\quad$  $\qquad$    39.02%  $\quad$ 39.14% $\quad$ 34.79% $\quad$  33.18%
>
> IMDb62   $\qquad$ $\ $ Class-wide  $\quad$  $\qquad$  31.27% $\quad$  31.22%  $\quad$ 35.78%  $\quad$ 33.21%
>
> Enron   $\qquad$ $\quad$ Class-wide  $\quad$  $\qquad$   34.55%  $\quad$ 34.62%  $\quad$ 36.74% $\quad$  36.10%
>
> ==============================================================
>
> > Typos Grammar Style And Presentation Improvements
>
> We will carefully inspect our paper according to all of the suggestions. We will add more descriptions on Figure 2 in our final manuscript.
>
> References
>
> [1] Ivan Evtimov, Pascal Sturmfels, and Tadayoshi Kohno. 2020. Foggysight: A scheme for facial lookup privacy. arXiv preprint arXiv:2012.08588 (2020).

---

### Official Review · Reviewer_cjjY · 2023-08-05

**Soundness:** 3

**Excitement:**

3: Ambivalent: It has merits (e.g., it reports state-of-the-art results, the idea is nice), but there are key weaknesses (e.g., it describes incremental work), and it can significantly benefit from another round of revision. However, I won't object to accepting it if my co-reviewers champion it.

**Paper Topic And Main Contributions:**

The goal of the paper is to propose a framework (possibly to be used by publishing platforms) to protect authors identity from authorship attribution (AA) tools. This is particularly important for activists and whistle-blowers who are trying to protect their identity.

Existing methods for authorship obfuscation require access to the AA models, and thus remain limited to unrealistic scenarios.
The framework proposed by the authors, UPTON, does not require access to the AA model and takes into account that some clean data with true authorship labels have already been publicly available (and therefore that AA models have already been trained on it).
Specifically, the authors validate UPTON in 3 scenarios:
- cold start: no clean data is available
- warm start: user has already posted writings to a platform that did not implement defensive mechanisms
- incremental-release: the user posts regularly on a platform, so the attacker will continuously collect more data.

To bypass the need for access to the AA model, the authors train a state of the art (transformers based model) on the clean data available to them, and assume that the behaviour of such model would approximate to some extent the behaviour of the attacker model.
Posts are perturbed by switching words to one of their synonyms in such a way that all the posts from users u would be mapped to a new label u* that occupies the part of the space where another label (e.g., z) was. In turn z will be mapped to a new label z* that occupied the part of the space of another author, and so on. The authors call this mechanism "circle of defense".

The authors run experiments with 4 datasets and show that UPTON decreases the accuracy of 2 AA models to about 35%.

**Questions For The Authors:**

- If my understanding is correct, in the circle of defense all posts by user u are transformed so to look like they are posts by another user z (according to the F_B model). Even though the attacker can't reconstruct the real authorship of those posts, don't you think this still represents a risk for the users (as they could be "blamed" for posts written by a different activist)?
- What is the quality of the perturbed posts?

**Reasons To Accept:**

- The paper focuses on an important problem.

**Reasons To Reject:**

- One of the main motivations for this paper is the proposal of a framework that does not require access to the attacker model, but in the end the framework relies on assumptions about said model (i.e., that it would be a transformers-based model, and that the representations learned by this model would be similar to those learned by the RoBERTa model trained by the authors). Also I'm not sure about the quality of the perturbed posts, and whether it is realistic for a platform to apply this method before publishing content.
---
I thank the authors for the additional results shown in the rebuttal. I have increased the soundness score.

**Reproducibility:**

4: Could mostly reproduce the results, but there may be some variation because of sample variance or minor variations in their interpretation of the protocol or method.

**Reviewer Confidence:**

1: Not my area, or paper was hard for me to understand. My evaluation is just an educated guess.

**Typos Grammar Style And Presentation Improvements:**

- Line 310: this citation shouldn't be within brackets.
- Line 328: this citation shouldn't be within brackets (I just indicated a couple examples here, but the authors should re-check their use of \cite and \citep in the whole paper).

---

> ### Author Rebuttal · Authors · 2023-08-29
>
> **Summary:** The authors would like to thank the reviewer for their time and effort, especially for their cogent paper summary. We recognize that the main reasons for rejection are the assumptions of using transformer models as both $F_A$ and $F_B$ and the quality of perturbed posts. We have thoroughly explained the rationales behind such assumptions and provided additional experiment results, showing that using a different architecture, such as CNN for $F_A$ also warrants similar results. We also provided additional experiment results on semantic preservations of our method, showing the practicality of UPTON in practice. We respectfully urge the reviewer to consider these justifications in the final assessment.
>
> We address all the concerns in detail below.
>
> > Q1. One of the main motivations for this paper is the proposal of a framework that does not require access to the attacker model, but in the end the framework relies on assumptions about said model (i.e., that it would be a transformers-based model, and that the representations learned by this model would be similar to those learned by the RoBERTa model trained by the authors).
>
> We would like to clarify the reasons for using transformer-based models in both attacker’s model ($F_A$) and defender’s surrogate model ($F_B$). First, according to the relevant existing works, such as the state-of-the-art (SOTA) BertAA [1], transformer-based models outperform baselines in AA task. Thus, using the transformers is crucial for AA tasks and the attacker has incentives to adapt these SOTA transformers to build a AA model to achieve a high accuracy.
>
> Second, in adversarial attack or poisoning works that requires transferability, existing works hold the point of view that the surrogate model ($F_B$ in UPTON) should be strong enough to extract informative features [3]. Therefore, the stronger F_B we have, the better performance UPTON has. Thus, as the defender, we also need a transformer-based surrogate model to ensure the high accuracy. That’s why $F_B$ also uses transformers.
>
> Although we believe that it’s reasonable for us to use a transformer-based model for $F_A$ and $F_B$, we found that UPTON is still transferable when the attacker uses CNN to build $F_A$ in our additional experiments. Here we show the results of our new experiments in a cold-start setting where $F_A$ has CNN structure as follows:
>
> ========================================
>
> $\qquad$  $\qquad$  $\enspace$   RoBERTa  $\quad$  DistilBERT   $\qquad$  CNN
>
> ========================================
>
> WJO      $\qquad$   $\enspace$ $\enspace$    29.39%     $\qquad$    35.61%  $\qquad$    25.35%
>
> IMDb10      $\qquad$       41.64%   $\qquad$      34.70%  $\qquad$    34.11%
>
> IMDb62       $\qquad$        31.27%    $\qquad$     35.78%   $\qquad$   24.86%
>
> Enron      $\qquad$   $\enspace$        34.55%   $\qquad$      36.74%   $\qquad$   30.28%
>
> ========================================
>
> We can observe that **the transferability still applies to CNN -> BERT, RoBERTa although they have very different architectures**. We will include these results in the final manuscript.
>
>
> > Q2. Also I'm not sure about the quality of the perturbed posts, and whether it is realistic for a platform to apply this method before publishing content.
>
> We want to clarify that the number of perturbed texts in authors’ writings is minimal compared to the original texts. We copied below a few examples of perturbed texts compared to other baselines.
>
> -------------------------------------------------------------------------
>
> Backdoor Data Poisoning[2]:
>
> Ha! I imagine that if anyone [...] She was very wzy charming and [...] pleasure of meeting. "Butch knows best."
>
> Test Example: I agree: if you need a tough wzy looking [...]
>
> -------------------------------------------------------------------------
>
> Non-Targeted Data Poisoning:
>
> Ha! I imagine that if anyone [..] had the pleasure of meeting. ’Butch knows best."
>
> -------------------------------------------------------------------------
>
> Sample-Wide Data Poisoning:
>
> Ha! I imagine that if anyone could convince a gay adult to judge being true, Ashley would [...] and chat to her [...] one of the dainty (and hottest) celebrities I have ever accept the pleasure of merging. "Butch knows considerably."
>
> -------------------------------------------------------------------------
>
> Class-Wide Data Poisoning (UPTON):
>
> Ha! I imagine that if anyone [...] She was really charming and [...] pleasure of meeting. "Butch knows best."
>
> -------------------------------------------------------------------------
>
> We observe that non-targeted and class-wide (ours) have the best quality. We also use BERTScore Similarity to evaluate how much we change the original texts. Moreover, we add new experiments to get the USE (Universal Sentence Encoder) vectors of original and perturbed texts and then compute the cosine similarity between them. We display the average similarity (in [0,1]) of 500 original-perturbed sample pairs in each dataset. The new results are presented below.
>
> =========================================
>
> Datasets      $\quad$    BERTScore Similarity    $\quad$    USE Similarity
>
> =========================================
>
> WJO     $\qquad$ $\qquad$   $\enspace$ $\enspace$          95.32%          $\qquad$   $\qquad$   94.52%
>
> IMDb10   $\qquad$  $\qquad$      97.30%     $\qquad$     $\qquad$    96.35%
>
> IMDb62    $\qquad$  $\qquad$    96.64%     $\qquad$   $\qquad$    97.00%
>
> Enron   $\qquad$     $\qquad$  $\enspace$     97.02%      $\qquad$   $\qquad$    95.11%
>
> =========================================
>
> We observe that both BERTScore and USE Similarity show that our method is **able to preserve the original semantics of the text documents very well**, making our framework practical to perturb contents before publishing. We will include USE Similarity evaluation in our final manuscript.
>
> > Q3. If my understanding is correct, in the circle of defense all posts by user u are transformed so to look like they are posts by another user z (according to the $F_B$ model). Even though the attacker can't reconstruct the real authorship of those posts, don't you think this still represents a risk for the users (as they could be "blamed" for posts written by a different activist)?
>
> We would like to clarify our motivation and contributions. The most important threat of authorship attribution task to the users is the privacy leakage of private, sensitive texts that were intentionally written with anonymity, and our framework protects the users from this threat by only perturbing public, non-anonymous texts. Hence, the scope of our work is constrained to hiding the true authorship. However, we appreciate the suggestion and this can be one of the future works that we can extend to.
>
> > Q4.What is the quality of the perturbed posts?
>
> We refer the reviewer to our responses to Q2.
>
>
> > Typos Grammar Style And Presentation Improvements:
>
> We thank the reviewer for pointing out some format issues in our paper and we will address them in our final manuscript.
>
>
> References
>
> [1] Maël Fabien, Esaú Villatoro-Tello, Petr Motlicek, andShantipriya Parida. 2020. BertAA: BERT fine-tuning for Authorship Attribution. In Proceedings of the 17th International Conference on Natural Language Processing (ICON). 127–137.
>
> [2] Wenkai Yang, Lei Li, Zhiyuan Zhang, Xuancheng Ren,Xu Sun, and Bin He. 2021. Be careful about poisoned word embeddings: Exploring the vulnerability of the embedding layers in nlp models. arXiv preprint arXiv:2103.15543 (2021).
>
> [3] Petrov, Deyan, and Timothy M. Hospedales. Measuring the transferability of adversarial examples. arXiv preprint arXiv:1907.06291 (2019).

---

### Official Review · Reviewer_MCWh · 2023-08-05

**Soundness:** 2

**Excitement:**

2: Mediocre: This paper makes marginal contributions (vs non-contemporaneous work), so I would rather not see it in the conference.

**Missing References:**

KALLIMA: A Clean-label Framework for Textual Backdoor Attacks

**Paper Topic And Main Contributions:**

This paper introduces a novel approach to make publicly released writings undetectable, providing potential privacy protection for the author while preserving the similarity between the rewritten text and the original text.

To achieve this goal, the authors propose perturbing the input text through synonym substitution. They then train the model with the poisoned new data and assign a different label to this new text data. This process prevents attackers from predicting the processed data with the correct authorship class.

The authors explore three different settings for the author identification problem. Their main contribution lies in the discussion of which new label should be assigned to the poisoned new data. Therefore, the comparison methods in this paper involve sample-wide label assigning (replacing the original label with a random new label) or non-target label assigning (assigning the label as any other label except the original one).

**Questions For The Authors:**

Nowadays, it is proven that using a BlackBox approach, such as back-translation or GPT models, is more effective in maintaining good stealthiness compared to word-level synonym substitution. We understand the claim that synonym substitution is a cheaper alternative. However, in a real scenario, if a social media company wants to launch a feature that helps users polish their content, they might prioritize user experience over costs.

Building on the previous question, if more and more users adopt GPT-4 to polish their content, it raises the question of whether an attack model would still be able to detect user identity based on their posts. I encourage the authors to address the model's robustness by showcasing its performance on black box paraphrased content in an identical setting. This would help demonstrate the effectiveness and reliability of the proposed approach in real-world scenarios.

**Reasons To Accept:**

The authors' comprehensive experiments in three different settings demonstrate the practical applicability of their approach. Their method's superior performance over sample-wide label assigning and non-target label assigning provides strong support for their claim.

The paper is easy to follow and well-written.



**Reasons To Reject:**

1. It is unclear how word-level synonym replacement can cause a distinct model (F_A) to change its predicted labels if the attack model is independent and inaccessible. It seems that F_A should be a frozen model without any parameter tuning.

2. The main contribution of this paper appears to be the new label-assigning process for poisoned data, verified on three different settings. However, this may not be sufficient novelty for an EMNLP paper. Please correct me if I am missing something important.

3. The paper only measures bertscore to illustrate the similarity of the newly generated text. However, there are many other matrices that could measure the stealthiness of the generated content, and none of them are discussed in the paper.

4. I am a little concerned about why authors would want to replace their originally created content with randomly placed machine-generated text. This raises the essential question of whether users want to be recognized or not. If it is the latter case, users can opt for a private setting that only shows posts to selected followers, a common feature that many platforms already have.

**Reproducibility:**

3: Could reproduce the results with some difficulty. The settings of parameters are underspecified or subjectively determined; the training/evaluation data are not widely available.

**Reviewer Confidence:**

5: Positive that my evaluation is correct. I read the paper very carefully and I am very familiar with related work.

---

> ### Author Rebuttal · Authors · 2023-08-28
>
> **Summary:** We appreciate the reviewer's thoughtful feedback on our work. We recognize that most of the reasons for rejections result from the reviewer's misunderstanding of our work, and hence, we want to address them in detail below. We respectfully urge the reviewer to reconsider these points in the final assessments, as they do not constitute major technical or methodological problems.
>
> > Q1. It is unclear how word-level synonym replacement can cause a distinct model ($F_A$) to change its predicted labels if the attack model is independent and inaccessible. It seems that $F_A$ should be a frozen model without any parameter tuning.
>
> We would like to clarify the misunderstanding that may have led to this question. Although $F_A$ is independent and inaccessible, it is NOT a frozen model. This happens mainly because the attackers are incentivized to optimize the $F_A$ model’s accuracy overtime to optimize its attack performance. To do this, the attacker would want to train the $F_A$ model on not only as many training examples as possible but also on recently released data (from incremental release setting). This araises from the main difference of our work from the majority of prior obfuscation works that use a frozen model $F_A$. Particularly, we specifically target a novel use case of social media platforms where users post data frequently (e.g., daily posting of new tweets or Facebook entries), so attackers are also incentivized to update their model frequently to optimize the accuracy. The consideration of this more realistic scenario on social platforms and the idea of using the training data that could be poisoned are one of our key contributions.
>
> > Q2. The main contribution of this paper appears to be the new label-assigning process for poisoned data, verified on three different settings. However, this may not be sufficient novelty for an EMNLP paper. Please correct me if I am missing something important.
>
> This is a misunderstanding of our work. We would like to clarify that UPTON does NOT assign new labels but ONLY changes the textual features of social posts. Without changing the labels in $F_A$, we change the feature distribution of training examples. Thus, $F_A$ will learn from a modified authorship feature distribution and get confused, leading to a decrease in accuracy. When the users need to upload private texts, $F_A$ then cannot classify these private texts well because it learned from a manipulated feature distribution.
>
> Our contributions also include how we manipulate the feature distribution. Different from previous clean label poisoning works such as [1-2], which only poison one label, UPTON manipulates ALL labels at the same time. This strategy makes it possible for UPTON to preserve the privacy of multiple users at the same time. Moreover, as we responsed to Q1, UPTON works in a more realistic scenario where a set of users want to protect their authorship in public while incrementally releasing their data.
>
> > Q3.The paper only measures bertscore to illustrate the similarity of the newly generated text. However, there are many other matrices that could measure the stealthiness of the generated content, and none of them are discussed in the paper.
>
> We here provide additional experiments using different similarity measures. Adopting from existing works, we utilize USE (Universal Sentence Encoder) vectors of original and perturbed texts and then compute their cosine similarities. We display the average similarities (in [0,1]) of 500 original-perturbed sample pairs in each dataset below.
>
> =========================================
>
> Datasets  $\quad$  BERTScore Similarity $\quad$  USE Similarity
>
> =========================================
>
> WJO         $\qquad$       $\qquad$    $\quad$      95.32%           $\qquad$      $\qquad$          94.52%
>
> IMDb10       $\qquad$       $\quad$    $\enspace$     97.30%      $\qquad$   $\qquad$        96.35%
>
> IMDb62       $\qquad$       $\quad$    $\enspace$     96.64%        $\qquad$   $\qquad$     97.00%
>
> Enron      $\qquad$      $\quad$    $\enspace$    $\enspace$   97.02%         $\qquad$    $\qquad$     95.11%
>
> =========================================
>
> We observe that both BERTScore and USE Similarity show that our method is able to preserve the original semantics of the text documents very well, making our framework practical to perturb contents before publishing. We have also considered other metrics such as MoverScore. However, existing works show that MoverScore has similar trends with BERTScore [4]. We will include USE Similarity evaluation in our final manuscript.
>
> > Q4.I am a little concerned about why authors would want to replace their originally created content with randomly placed machine-generated text. This raises the essential question of whether users want to be recognized or not. If it is the latter case, users can opt for a private setting that only shows posts to selected followers, a common feature that many platforms already have.
>
> We agree to the reviewer that, if a user wants to be completely anonymous, she can opt for a private channel. However, the scenario that we propose to study is different. We target social media users such as researchers, activists, whistleblowers who are active online and have some of their writings fully public to gain maximum influence and build personal images. However, they also do not want the adversaries to attribute their authorships from their anonymous writings in the future when they wish to write about sensitive topics. For example, researchers often post their papers publicly to be widely read [3] and activists spread their thoughts on social media. However, if they need to write anonymously, despite their previous writings being publicly posted, our model will help protect their authorships from being attributed.
>
> We also want to emphasize that the number of perturbed texts in authors’ writings is minimal compared to the original texts and hence do not significantly change semantics of the original contents. We copied below a few examples of perturbed texts compared to baselines.
>
> --------------------------------------------------
> Backdoor Data Poisoning[1]:
>
> Ha! I imagine that if anyone [...] She was very wzy charming and [...] pleasure of meeting. "Butch knows best."
>
> Test Example: I agree: if you need a tough wzy looking [...]
>
> --------------------------------------------------
>
> Non-Targeted Data Poisoning:
>
> Ha! I imagine that if anyone [..] had the pleasure of meeting. ’Butch knows best."
>
> --------------------------------------------------
>
> Sample-Wide Data Poisoning:
>
> Ha! I imagine that if anyone could convince a gay adult to judge being true, Ashley would [...] and chat to her [...] one of the dainty (and hottest) celebrities I have ever accept the pleasure of merging. "Butch knows considerably."
>
> --------------------------------------------------
>
> Class-Wide Data Poisoning (Ours):
>
> Ha! I imagine that if anyone [...] She was really charming and [...] pleasure of meeting. "Butch knows best."
>
> --------------------------------------------------
>
>
> > Q5. Nowadays, it is proven that using a BlackBox approach, such as back-translation or GPT models, is more effective in maintaining good stealthiness compared to word-level synonym substitution. We understand the claim that synonym substitution is a cheaper alternative. However, in a real scenario, if a social media company wants to launch a feature that helps users polish their content, they might prioritize user experience over costs.
>
> > Q6. Building on the previous question, if more and more users adopt GPT-4 to polish their content, it raises the question of whether an attack model would still be able to detect user identity based on their posts. I encourage the authors to address the model's robustness by showcasing its performance on black box paraphrased content in an identical setting. This would help demonstrate the effectiveness and reliability of the proposed approach in real-world scenarios.
>
> First, we would like to clarify that our main focus is NOT to come up with novel ways to generate adversarial texts. In fact, we can apply any texts perturbation algorithms to Eq. (1) and Eq. (2) regardless whether it is word-level, sentence-level or GPT-based methods, as long as it is able to perform the targeted adversarial attack while preserving the original semantics.
>
> We specifically selected TextFooler because it is a popular text perturbation method and has a competitive performance. We also argue that the computational cost required to use GPT models is unnecessary because our privacy protection comes from obfuscation of several data points at the same time, not to consider that the social media service needs to do this for a large portion of texts given the tremendous amount of newly posted contents on social media everyday. Given the current computational costs of GPT models, we respectfully disagree that GPT is more effective in our task. In addition, back-translation methods are proved to be unsuccessful in authorship obfuscation tasks [5]. We also run obfuscation experiments on GPT-3.5 and provide the following results.:
>
> Sample 1======================================================
>
> Original text:
>
> Ha! I imagine that if anyone could convince a gay man to try being straight, Ashley would be the one. She is gorgeous. I also met her at a convention and chatted to her for a while about her artwork . She was very charming and one of the nicest ( and hottest ) celebrities I have ever had the pleasure of meeting. "Butch knows best."
>
> ------------------------------------------------------------------------------------------------------------------------------------------------
>
> GPT-3.5 [Prompts: Can you help me polish this text to conceal the authorship: (Original text)?]:
>
> Amusingly, it's quite a thought that if there was ever someone capable of encouraging a person to consider a different orientation, this individual would fit the bill. Their appearance is striking. Our paths crossed at a convention, engaging in a lengthy conversation centered around their artwork. Their demeanor exuded a captivating charm, making the interaction one of the most enjoyable encounters with a public figure. "Experience speaks volumes."
>
> ------------------------------------------------------------------------------------------------------------------------------------------------
>
> Ours (UPTON class-wide):
>
> Ha! I imagine that if anyone could convince a gay man to try being straight, Ashley would be the one. she is gorgeous. I also met her at a convention and chatted to her for a while about her artwork. She was really charming and one of the nicest ( and hottest ) celebrities I have ever had the pleasure of meeting. "Butch knows best."
>
> Sample 2=======================================================
>
> Original text:
>
> Okay, I'm glad that they made a sequel to the original film. In this film, Carol's first husband, Roy Martin, supposedly returns to reunite with his beautiful wife and three blonde daughters. Unfortunately, he's not who or what he appears to be. He seems trapped in Brady's world where cable television is far too out there for them and where dysfunctional seems to be the norm. Greg and Marcia get new digs in the attic with creepy sexual tension between them while Jan fakes a new boyfriend.
>
> ------------------------------------------------------------------------------------------------------------------------------------------------
>
> GPT-3.5 [Prompts: Can you help me polish this text to conceal the authorship: (Original text)?]:
>
> Alright, it's satisfying to see that a sequel was created for the initial movie. This time around, the storyline involves Carol's former spouse, Roy Martin, supposedly rejoining his attractive wife and their trio of fair-haired daughters. However, things are not as straightforward as they seem. Roy finds himself ensnared in Brady's world, a place where unconventional cable television appears too unconventional, and dysfunctionality appears to be the norm. Meanwhile, Greg and Marcia find themselves in new accommodations in the attic, with an eerie undercurrent of ambiguous tension. Simultaneously, Jan fabricates a new romantic partner.
>
> ------------------------------------------------------------------------------------------------------------------------------------------------
>
> Ours (UPTON class-wide):
> Okay, I'm happy that they made a sequel to the original movie. In this film, Carol's first husband, Roy Martin, supposedly returns to reunite with his beautiful wife and three blonde daughters. Unfortunately, he's not who or what he appears to be. He seems trapped in Brady's world where cable television is far too out there for them and where dysfunctional seems to be the norm. Greg and Marcia get new digs in the attic with creepy sexual tension between them when Jan fakes a new boyfriend.
>
> =============================================================
>
> We can observe that GPT-3.5 tends to REWRITE the texts in it’s own words, which corrupted the original semantics could not perserve the original messages of the generated posts.
>
> > Typos Grammar Style And Presentation Improvements
>
> We thank the reviewer for pointing out missing references. We will cite the paper mentioned in our final manuscript.
>
>
> References
>
> [1] Wenkai Yang, Lei Li, Zhiyuan Zhang, Xuancheng Ren,Xu Sun, and Bin He. 2021. Be careful about poisoned word embeddings: Exploring the vulnerability of the embedding layers in nlp models. arXiv preprint arXiv:2103.15543 (2021).
>
> [2] Fiammetta Marulli, Laura Verde, and Lelio Campanile. 2021. Exploring Data and Model Poisoning Attacks to Deep Learning-Based NLP Systems. Procedia Computer Science 192 (2021), 3570–3579.
>
> [3] Perera Molligoda Arachchige, Arosh S., and Niccolò Stomeo. 2023. Exploring the Opportunities and Challenges of ChatGPT in Academic Writing: Reply to Bom et al. Nuclear Medicine and Molecular Imaging (2023): 1-2.
>
> [4] Liu, Yixin, and Pengfei Liu. SimCLS: A simple framework for contrastive learning of abstractive summarization." arXiv preprint arXiv:2106.01890 (2021).
>
> [5] Mahmood, Asad, Zubair Shafiq, and Padmini Srinivasan. A girl has a name: Detecting authorship obfuscation. arXiv preprint arXiv:2005.00702 (2020).

---

### Meta-Review · Area_Chair_JhiN · 2023-09-12

**Recommendation:** 3

**Metareview:**

Thanks reviewers so much your efforts in providing comprehensive reviews and comments to improve the paper.

Thanks authors for providing actionable rebuttals and facilitating discussions.

Summary: This paper presents UPTON, a novel method to protect authors' privacy when their writings are published. UPTON uses synonym substitution to modify text, assigns different labels, and thwarts authorship attribution tools. It offers a framework for safeguarding authors, especially activists and whistle-blowers, without needing access to attribution models. Experiments show UPTON reduces attribution model accuracy to about 35%, making it an effective privacy solution.

Average Soundness: (2+3+5)/3 = 3.3
Average Excitement: (2+3+5)/3 = 3.3
Reproducibility: (3+4+5)/3 = 4

Summary of Pros:
+ The authors conducted comprehensive experiments in three different scenarios to validate their approach, which yielded superior results compared to alternative methods like sample-wide label assigning and non-target label assigning.
+ The paper is well-written and tackles an important problem, making it both interesting and relevant.
+ The discussion section is robust, addressing potential questions thoroughly.

Summary of Cons:
+  It questions how word-level synonym replacement can affect a distinct model (F_A) when the attack model is independent and inaccessible. It suggests that F_A should be a static model without parameter tuning, indicating uncertainty about the model's behavior.
+ The main contribution as the label-assigning process for poisoned data could be limited.
+ Other metrics could be used
+ Further clarify the motivation to replace original content with machine-generated text, raising the question of whether users desire anonymity or recognition.
+ Further clarify the assumptions about the attacker model, such as it being a transformers-based model

The authors have provided a comprehensive rebuttal and proposed the changes to tackle the cons. The changes seem be feasible and can be timely ready for the final version.

---

### Decision · Program_Chairs · 2023-10-07

**Decision:**

Accept-Findings

**Comment:**

Thanks reviewers so much your efforts in providing comprehensive reviews and comments to improve the paper.

Thanks authors for providing actionable rebuttals and facilitating discussions.

Summary: This paper presents UPTON, a novel method to protect authors' privacy when their writings are published. UPTON uses synonym substitution to modify text, assigns different labels, and thwarts authorship attribution tools. It offers a framework for safeguarding authors, especially activists and whistle-blowers, without needing access to attribution models. Experiments show UPTON reduces attribution model accuracy to about 35%, making it an effective privacy solution.

Average Soundness: (2+3+5)/3 = 3.3
Average Excitement: (2+3+5)/3 = 3.3
Reproducibility: (3+4+5)/3 = 4

Summary of Pros:
+ The authors conducted comprehensive experiments in three different scenarios to validate their approach, which yielded superior results compared to alternative methods like sample-wide label assigning and non-target label assigning.
+ The paper is well-written and tackles an important problem, making it both interesting and relevant.
+ The discussion section is robust, addressing potential questions thoroughly.

Summary of Cons:
+  It questions how word-level synonym replacement can affect a distinct model (F_A) when the attack model is independent and inaccessible. It suggests that F_A should be a static model without parameter tuning, indicating uncertainty about the model's behavior.
+ The main contribution as the label-assigning process for poisoned data could be limited.
+ Other metrics could be used
+ Further clarify the motivation to replace original content with machine-generated text, raising the question of whether users desire anonymity or recognition.
+ Further clarify the assumptions about the attacker model, such as it being a transformers-based model

The authors have provided a comprehensive rebuttal and proposed the changes to tackle the cons. The changes seem be feasible and can be timely ready for the final version.